# Do people with symptoms of an infectious illness follow advice to stay at home? Evidence from a series of cross-sectional surveys about presenteeism in the UK

G James Rubin,[1] Louise E Smith [iD],[1] Richard Amlot,[2] Nicola T Fear [iD],[3] Henry Potts,[4] Susan Michie[5]

GJR and LES contributed equally.

GJR and LES are joint first authors.

For numbered affiliations see end of article.

**Correspondence to**
Dr G James Rubin;
Gideon.rubin@kcl.ac.uk

## ABSTRACT

**Objectives** To assess the percentage of people in the UK with cough, fever or loss of taste or smell who have not had a positive COVID-19 test result who had been to work, to shops, socialised or provided care to a vulnerable person in the 10 days after developing symptoms. To investigate whether these rates differed according to the type of symptom, what the participant thought the cause of their symptoms was and whether they had taken a COVID-19 test.

**Design** Four online cross-sectional surveys using non-probability quota sampling method (n=8547).

**Setting** Data were collected across the UK from 20 September to 3 November 2021, via a market research company.

**Participants** Aged over 16 years living in the UK.

**Primary outcome measures** Out-of-home activity.

**Results** 498 participants reported one or more symptoms and had not had a positive COVID-19 test result. Within that group, about half of employed participants had attended work while symptomatic (51.2%–56.3% depending on the symptom, 95% CIs 42.2% to 65.6%). Rates of other contact behaviours ranged from 31.4% (caring for a vulnerable person after developing a cough: 95% CI 24.3% to 38.4%) to 61.5% (shopping for groceries or pharmacy after developing a cough: 95% CI 54.1% to 68.9%). There were no differences according to type of symptom experienced or what the participant felt might be the cause. People who had taken a COVID-19 test were less likely to go out shopping for non-essentials than people who had not taken a test.

**Conclusion** Many people in the UK with symptoms of an infectious disease were not following government advice to stay at home if they believed they had an infectious illness. Reducing these rates may require a shift in our national attitude to the acceptability of people attending work with infectious illnesses.

## INTRODUCTION

The spread of infectious disease within a workplace represents a risk to employees, those who come into contact with them

## STRENGTHS AND LIMITATIONS OF THIS STUDY

⇒ The use of a large (n=8547) sample allowed identification of a large number of people who had recently experienced symptoms (n=548).
⇒ By asking about symptoms and activities that had occurred in the past 10 days, we limited the likelihood of recall bias.
⇒ The sample was derived using a non-probability quota sampling method, hence the representativeness of the sample is unclear.
⇒ The data are self-reported and therefore, given that it may be socially undesirable to admit attending work while ill, our reported rate of people attending work while symptomatic may be an underestimate.

and to the productivity of the organisation. Although transmission can be reduced if employees who are ill stay at home, systematic reviews have demonstrated that there are a wide range of factors that determine whether someone will have the capability, opportunity or motivation to do so.[1 2]

The COVID-19 pandemic has made this issue particularly pressing. In the UK, at the time these data were collected, people who developed a new continuous cough, a fever or a loss or change to their sense of taste or smell (the then so-called 'cardinal symptoms' of COVID-19) were urged to stay at home and to take a PCR test. If they received a positive test result, they were then legally obliged to remain at home. Yet, self-isolating if you receive a positive COVID-19 result was only one aspect of the UK government's plan to ease pressure on the National Health Service during the pandemic. According to government guidance at the time, people should also have '(tried) to stay at home if you are feeling unwell' even in the absence of a positive COVID-19 test, while businesses were

'encouraged to ask employees to stay at home if they are feeling unwell'.[3] This was intended to reduce the spread of influenza-like illness throughout workplaces and reinforced pre-existing advice to stay off work or school when ill.[4]

While a growing body of research has investigated the extent to which people adhere to COVID-19 testing and self-isolation policies,[5] we are unaware of evidence about the extent to which people adhere to advice to 'stay at home if you are feeling unwell' even in the absence of a positive COVID-19 diagnosis. In this study, we investigated the behaviour of people who reported having symptoms, but who did not have a positive COVID-19 test result, using a series of national surveys. We analysed rates of five behaviours among people who were symptomatic: going to work, going to the shops for food or medicine, going to the shops for other things, meeting up with friends or family you do not live with and providing help or care for a vulnerable person. We tested whether rates differed according to the symptom that was reported, what the participant believed might have caused their symptoms, and whether the participant had taken a test for COVID-19. We also assessed associations between behaviours and sociodemographic variables.

## METHODS
### Design
The English Department of Health and Social Care commissioned a series of cross-sectional, nationally representative surveys from market research companies, beginning in January 2020. For this analysis, we used data from survey waves 58–61 (20 September to 3 November 2021). More details are available elsewhere.[6 7]

### Participants
Participants were aged over 16 years, lived in the UK and had previously opted-in to receive invitations to take part in online market research surveys. Participants who took part in one wave of the Department of Health and Social Care series of surveys were prevented from taking part in any of the next three waves. We therefore elected to use the most recent four waves to ensure that each participant only contributed one set of responses. The market research company sent invitations to existing panel members who met these criteria, asking them to complete the survey. Quotas were used based on age and gender (combined) and region to ensure the sample was broadly representative of the UK population on these variables. Quotas were based on mid-year (2018) projections from the Office for National Statistics.

### Questionnaire items
Participants were asked whether they had developed any from a list of 13 symptoms in the past 10 days. This included new continuous cough, high temperature/fever, loss of sense of smell and loss of taste. Participants who reported one of these symptoms were asked 'what do

you think your symptoms could have been caused by' and were able to tick one or more of: 'hayfever/allergies', 'a cold', 'asthma', 'influenza', 'coronavirus', 'long COVID-19', 'other, please specify' (with a free-text space provided) and 'don't know'. They were also asked which, if any, of a list of actions they took 'while you had these symptoms', which included 'I took a test to confirm whether I have coronavirus'. Anyone who ticked this option was asked whether they took a PCR test or a lateral flow test (LFT). Participants reporting cardinal symptoms were also asked 'how many times after developing symptoms recently, if at all, have you completed each of these activities' and were presented with a list of 11 daily activities, of which 5 might involve contact with other people.

We asked participants for their age, gender, whether there were dependent children in their household, employment status (for those in full-time employment, part-time employment or self-employment, we also asked whether they could work from home), socioeconomic grade, ethnicity, first language, how many people lived in their household, vaccination status and whether they thought or had confirmed SARS-CoV-2 infection previously. Participants were also asked three questions about their financial hardship in the last week (to what extent they had been struggling to make ends meet, skipping meals they would usually have and were finding their current living situation difficult; Cronbach's $\alpha=0.76$); these items were summed to make a continuous scale, with a higher score indicated greater hardship. Participants' region was derived from their postcode.

### Patient and public involvement
Lay members served on the advisory group for the project that developed our prototype survey material; this included three rounds of qualitative testing.[8] Due to the rapid nature of this research, the public was not involved in the further development of the materials during the COVID-19 pandemic.

### Analysis
We restricted our sample to participants who reported having experienced one or more cardinal symptoms. We excluded anyone who reported that their most recent test for COVID-19 was positive, on the basis that such people would be legally obliged to self-isolate and should have received support and regular encouragement to do so via the UK's NHS Test and Trace service. Our sample therefore consisted of people who had received a negative test result and people who had not taken a test.

We calculated the percentage of participants who reported engaging in each of the five activities (going out to the shops for groceries/pharmacy, to shops for things other than groceries/pharmacy, to work, to meet friends or family from another household and to provide help or care for a vulnerable person). We restricted analyses of work attendance to those who reported being in full-time, part-time or self-employment.

Logistic regressions (engaged in this activity, vs did not engage in this activity) were used to assess associations between activities and sociodemographic variables (survey wave, region, gender, age (raw and quadratic), ethnicity, socioeconomic grade, index of multiple deprivation, presence of dependent children in the household, employment status, education, English as first language, perceived immunity (composite score: been vaccinated or think had COVID-19, vs not been vaccinated and think have not had COVID-19), living alone, financial hardship).

We used logistic regressions to assess whether behaviour was associated with symptoms, attribution of symptoms (by grouping into mutually exclusive groups according to whether they reported COVID-19 as a possible cause, whether they reported another infectious disease as a possible cause (excluding COVID-19), or whether they did not list any infectious disease as a cause), and whether participants had sought a test for COVID-19 because of their recent symptoms separately. A second set of logistic regression analyses were conducted, adjusting for sociodemographic variables.

Due to the large number of analyses conducted on a single outcome (n=23), we implemented a Bonferroni correction (p≤0.002) for reporting results as being statistically significant, but we give raw p values in the paper for completeness.

## RESULTS

Out of a total sample of 8547, 548 participants (6.4%) reported cardinal symptoms. Of those, 50 reported that their most recent COVID-19 test was positive and were excluded. Of the remaining 498: 54.8% (n=273) were male, 44.8% (n=223) were female and 0.4% (n=2) preferred to self-describe or not say; 71.3% (n=355) identified as white British, 7.4% (n=37) identified as white other, 20.1% (n=100) identified as being from another ethnic group and 1.2% (n=6) preferred not to say; and 71.5% (n=365) were working, 26.7% (n=133) were not working and 1.8% (n=9) preferred not to say. The mean age was 36.3 years (SD 15.2 years).

About half of employed people with fever (52.5%, 95% CI 43.5% to 61.4%), cough (51.2%, 95% CI 42.2% to 60.3%) and loss of taste or smell (56.3%, 95% CI 46.9% to 65.6%) reported having been to work after developing their symptoms. Rates of going to the shops, meeting friends or family that you do not live with, and providing care for a vulnerable person ranged from 31.4% (95% CI 24.3% to 38.4%) to 61.5% (95% CI 54.1% to 68.9%; table 1).

Providing help or care for a vulnerable person was statistically significantly associated with having a dependent child, lesser financial hardship and survey wave, with fewer people leaving home for this reason in later waves. See online supplemental file 1 for full results.

Rates of out-of-home activity did not differ according to what symptom was reported or what the participant believed had caused their symptoms (table 2). Participants who had taken a test to check whether they had COVID-19 were generally less likely to engage in each of the behaviours aside from going to work in unadjusted analyses. In adjusted analyses, only the association between going out for items other than groceries/pharmacy and having taken a test remained significant when applying our Bonferroni correction.

## DISCUSSION

In the UK, while advice prior to the pandemic[4] and subsequent messages[3] asked people to stay at home if they felt unwell, our data show that about half of people with cough, fever or loss of their sense of taste or smell, and who did not have a positive COVID-19 test result, went to work, went shopping, socialised with others and provided care to vulnerable people.

Similar findings have been reported elsewhere. For example, one cross-sectional survey of Japanese workers identified 82 people who had experienced fever or cold symptoms between February and May 2020.[9] Of those, 51 (62.2%) reported having been to work within 7 days of symptom onset. Similarly, between 2006 and 2011, a community cohort study in England found that only a third of people with influenza-like illness reported having taken any time off work while ill.[10] This high level of 'presenteeism' is problematic. Not only is limiting the spread of infection within a workforce or community challenging if people continue to mix while symptomatic, but employees are also less likely to function effectively at work while ill.[11] Identifying ways to reduce presenteeism may pay dividends in terms of both health and economic outcomes.[12]

While it might be expected that such behaviours would occur less frequently in people who have a fever or who believe themselves to be infectious, we did not find this to be the case. Previous reviews have noted a wide range of factors that can prevent someone from staying at home when ill, including the absence of paid sick leave, organisational culture, a sense of professional obligation and concern about the impact of your absence on other people.[1 2] It may be that such factors are more relevant than knowledge as to whether you might be infectious in determining behaviour when ill. This has implications for interventions to reduce presenteeism. Encouragement to stay at home may be insufficient in the absence of support, change in organisational processes, increased sick pay and changes in our national attitudes, practices and policies regarding presenteeism.

Participants who had taken a test for COVID-19 were less likely than those who had not taken a test to have engaged in the five behaviours that we assessed in unadjusted analyses, although only the association between going out for items other than groceries/pharmacy and having taken a test remained significant after correcting for multiple comparisons. Given that attributing your symptoms to COVID-19 did not affect behaviour, why

**Table 1** Behaviours among people with fever, cough or loss of taste or smell and who have not had a positive COVID-19 test

| | | Been out to.... | | | | | | | | | | | | | | | | | | | | |
|---|---|---|---|---|---|---|---|---|---|---|---|---|---|---|---|---|---|---|---|---|---|---|
| | | The shops for groceries/pharmacy | | | | | | The shops for things other than groceries/pharmacy | | | | | | Go to work (total n=356) | | | | | | Meet friends or family that you don't live with | | | | | | Provide help or care for a vulnerable person | | | | | |
| | | No | | | Yes | | | No | | | Yes | | | No | | | Yes | | | No | | | Yes | | | No | | | Yes | | |
| | | % (95% CI) | n | | % (95% CI) | n | | % (95% CI) | n | | % (95% CI) | n | | % (95% CI) | n | | % (95% CI) | n | | % (95% CI) | n | | % (95% CI) | n | | % (95% CI) | n | | % (95% CI) | n | |
| Symptom | Fever | 40.7 (33.2 to 48.2) | 68 | | 59.3 (51.8 to 66.8) | 99 | | 48.5 (40.8 to 56.2) | 81 | | 51.5 (43.8 to 59.2) | 86 | | 47.5 (38.6 to 56.5) | 58 | | 52.5 (43.5 to 61.4) | 64 | | 50.9 (43.2 to 58.6) | 85 | | 49.1 (41.4 to 56.8) | 82 | | 59.9 (52.4 to 67.4) | 100 | | 40.1 (32.6 to 47.6) | 67 | |
| | Cough | 38.5 (31.1 to 45.9) | 65 | | 61.5 (54.1 to 68.9) | 104 | | 55.6 (48.1 to 63.2) | 94 | | 44.4 (36.8 to 51.9) | 75 | | 48.8 (39.7 to 57.8) | 59 | | 51.2 (42.2 to 60.3) | 59 | | 52.7 (45.1 to 60.3) | 89 | | 47.3 (39.7 to 54.9) | 80 | | 68.6 (61.6 to 75.7) | 116 | | 31.4 (24.3 to 38.4) | 53 | |
| | Loss of taste/ smell | 39.1 (31.5 to 46.8) | 63 | | 60.9 (53.2 to 68.5) | 98 | | 44.1 (36.3 to 51.9) | 71 | | 55.9 (48.1 to 63.7) | 90 | | 43.8 (34.4 to 53.1) | 49 | | 56.3 (46.9 to 65.6) | 63 | | 48.4 (40.6 to 56.2) | 78 | | 51.6 (43.8 to 59.4) | 83 | | 60.2 (52.6 to 67.9) | 97 | | 39.8 (32.1 to 47.4) | 64 | |
| Attributed to* | COVID-19 | 41.6 (31.8 to 51.4) | 42 | | 58.4 (48.6 to 68.2) | 59 | | 48.5 (38.6 to 58.4) | 49 | | 51.5 (41.6 to 61.4) | 52 | | 51.2 (40.2 to 62.3) | 42 | | 48.8 (37.7 to 59.8) | 40 | | 53.5 (43.6 to 63.4) | 54 | | 46.5 (36.6 to 56.4) | 47 | | 59.4 (49.7 to 69.1) | 60 | | 40.6 (30.9 to 50.3) | 41 | |
| | Infectious illness | 40.5 (35.0 to 46.0) | 125 | | 59.5 (54.0 to 65.0) | 184 | | 50.8 (45.2 to 56.4) | 157 | | 49.2 (43.6 to 54.8) | 152 | | 43.9 (37.1 to 50.6) | 93 | | 56.1 (49.4 to 62.9) | 93 | | 52.8 (47.2 to 58.3) | 163 | | 47.2 (41.7 to 52.8) | 146 | | 64.1 (58.7 to 69.5) | 198 | | 35.9 (30.5 to 41.3) | 111 | |
| | Non-infectious illness | 33.3 (23.2 to 43.4) | 29 | | 66.7 (56.6 to 76.8) | 58 | | 46.0 (35.3 to 56.7) | 40 | | 54.0 (43.3 to 64.7) | 47 | | 50.8 (37.9 to 63.7) | 31 | | 49.2 (36.3 to 62.1) | 31 | | 40.2 (29.7 to 50.7) | 35 | | 59.8 (49.3 to 70.3) | 52 | | 63.2 (52.9 to 73.6) | 55 | | 36.8 (26.4 to 47.1) | 32 | |
| Testing uptake* | No test | 33.8 (28.4 to 39.2) | 100 | | 66.2 (60.8 to 71.6) | 196 | | 40.9 (35.2 to 46.5) | 121 | | 59.1 (53.5 to 64.8) | 175 | | 41.2 (34.6 to 47.7) | 91 | | 58.8 (52.3 to 65.4) | 130 | | 44.3 (38.6 to 49.9) | 131 | | 55.7 (50.1 to 61.4) | 165 | | 54.4 (48.7 to 60.1) | 161 | | 45.6 (39.9 to 51.3) | 135 | |
| | LFT only | 40.2 (30 to 50.4) | 37 | | 59.8 (49.6 to 70) | 55 | | 60.9 (50.7 to 71) | 56 | | 39.1 (29 to 49.3) | 36 | | 55.2 (42.0 to 68.4) | 32 | | 44.8 (31.6 to 58) | 26 | | 58.7 (48.4 to 68.9) | 54 | | 41.3 (31.1 to 51.6) | 38 | | 81.5 (73.4 to 89.6) | 75 | | 18.5 (10.4 to 26.6) | 17 | |
| | PCR test | 54.1 (44.6 to 63.6) | 59 | | 45.9 (36.4 to 55.4) | 50 | | 63.3 (54.1 to 72.5) | 69 | | 36.7 (27.5 to 45.9) | 40 | | 56.6 (45.2 to 68) | 43 | | 43.4 (32 to 54.8) | 33 | | 61.5 (52.2 to 70.8) | 67 | | 38.5 (29.2 to 47.8) | 42 | | 70.6 (62.0 to 79.3) | 77 | | 29.4 (20.7 to 38) | 32 | |

*Due to a technical error, one person was not asked about their symptom attribution or test type, therefore n for those analyses is 497.
LFT, lateral flow test.

**Table 2** Associations between activity and symptom, symptom attribution and requesting a test

| | | Been out to… | | | | | | | | | | | | | | | | | | | |
|---|---|---|---|---|---|---|---|---|---|---|---|---|---|---|---|---|---|---|---|---|---|
| | | The shops for groceries/pharmacy | | | | The shops for things other than groceries/pharmacy | | | | Go to work | | | | Meet friends or family that you don't live with | | | | Provide help or care for a vulnerable person | | | |
| | | OR for going out (95% CI)† | P value | aOR for going out (95% CI)‡ § | P value | OR for going out (95% CI)† | P value | aOR for going out (95% CI)‡ § | P value | OR for going out (95% CI)¶ | P value | aOR for going out (95% CI)§** | P value | OR for going out (95% CI)† | P value | aOR for going out (95% CI)‡ § | P value | OR for going out (95% CI)† | P value | aOR for going out (95% CI)‡§ | P value |
| Symptom | Fever | Ref | – | Ref | – | Ref | – | Ref | – | Ref | – | Ref | – | Ref | – | Ref | – | Ref | – | Ref | – |
| | Cough | 1.10 (0.71 to 1.70) | 0.67 | 1.30 (0.80 to 2.13) | 0.29 | 0.75 (0.49 to 1.15) | 0.19 | 0.93 (0.57 to 1.54) | 0.79 | 0.95 (0.58 to 1.58) | 0.85 | 1.35 (0.73 to 2.51) | 0.34 | 0.93 (0.61 to 1.43) | 0.75 | 0.95 (0.58 to 1.56) | 0.85 | 0.68 (0.44 to 1.07) | 0.09 | 0.87 (0.50 to 1.53) | 0.64 |
| | Loss of taste/smell | 1.07 (0.69 to 1.66) | 0.77 | 1.03 (0.63 to 1.7) | 0.91 | 1.19 (0.77 to 1.84) | 0.42 | 1.12 (0.67 to 1.86) | 0.66 | 1.17 (0.70 to 1.95) | 0.56 | 1.10 (0.59 to 2.04) | 0.77 | 1.10 (0.72 to 1.70) | 0.66 | 0.99 (0.60 to 1.62) | 0.96 | 0.98 (0.63 to 1.53) | 0.95 | 0.80 (0.47 to 1.37) | 0.42 |
| | Overall | χ²(2)=0.2 | 0.91 | χ²(2)=1.3 | 0.52 | χ²(2)=4.5 | 0.11 | χ²(2)=0.5 | 0.78 | χ²(2)=0.6 | 0.73 | χ²(2)=1.0 | 0.62 | χ²(2)=0.6 | 0.74 | χ²(2)=0.0 | 0.98 | χ²(2)=3.5 | 0.17 | χ²(2)=0.7 | 0.72 |
| Attributed to† | COVID-19 | Ref | – | Ref | – | Ref | – | Ref | – | Ref | – | Ref | – | Ref | – | Ref | – | Ref | – | Ref | – |
| | Infectious illness | 1.05 (0.66 to 1.65) | 0.84 | 1.34 (0.80 to 2.26) | 0.27 | 0.91 (0.58 to 1.43) | 0.69 | 1.51 (0.90 to 2.55) | 0.12 | 1.34 (0.81 to 2.24) | 0.26 | 2.37 (1.26 to 4.45) | 0.008 | 1.03 (0.66 to 1.61) | 0.90 | 1.37 (0.82 to 2.31) | 0.23 | 0.82 (0.52 to 1.30) | 0.40 | 1.32 (0.76 to 2.30) | 0.32 |
| | Non-infectious illness | 1.42 (0.78 to 2.58) | 0.25 | 2.06 (1.05 to 4.05) | 0.04 | 1.11 (0.62 to 1.97) | 0.73 | 1.58 (0.81 to 3.07) | 0.18 | 1.02 (0.52 to 1.97) | 0.96 | 1.33 (0.59 to 3.02) | 0.49 | 1.71 (0.96 to 3.05) | 0.07 | 2.47 (1.27 to 4.80) | 0.008 | 0.85 (0.47 to 1.54) | 0.59 | 1.30 (0.64 to 2.63) | 0.47 |
| | Overall | χ²(2)=1.7 | 0.43 | χ²(2)=4.4 | 0.11 | χ²(2)=0.7 | 0.71 | χ²(2)=2.7 | 0.26 | χ²(2)=1.8 | 0.41 | χ²(2)=8.0 | 0.02 | χ²(2)=4.6 | 0.10 | χ²(2)=7.3 | 0.03 | χ²(2)=0.7 | 0.70 | χ²(2)=1.0 | 0.60 |
| Testing uptake† | No test | Ref | – | Ref | – | Ref | – | Ref | – | Ref | – | Ref | – | Ref | – | Ref | – | Ref | – | Ref | – |
| | LFT only | 0.76 (0.47 to 1.23) | 0.26 | 0.84 (0.48 to 1.46) | 0.53 | 0.44 (0.28 to 0.72) | 0.001* | 0.58 (0.33 to 1.02) | 0.06 | 0.57 (0.32 to 1.02) | 0.06 | 0.87 (0.42 to 1.78) | 0.70 | 0.56 (0.35 to 0.90) | 0.02 | 0.70 (0.40 to 1.22) | 0.20 | 0.27 (0.15 to 0.48) | <0.001* | 0.40 (0.20 to 0.80) | 0.01 |
| | PCR test | 0.43 (0.28 to 0.68) | <0.001* | 0.44 (0.27 to 0.74) | 0.002* | 0.4 (0.25 to 0.63) | <0.001* | 0.38 (0.22 to 0.65) | <0.001* | 0.54 (0.32 to 0.91) | 0.02 | 0.49 (0.25 to 0.96) | 0.04 | 0.50 (0.32 to 0.78) | 0.002* | 0.47 (0.28 to 0.80) | 0.005 | 0.50 (0.31 to 0.79) | 0.004 | 0.43 (0.24 to 0.78) | 0.005 |
| | Overall | χ²(2)=13.5 | 0.001* | χ²(2)=9.6 | 0.008 | χ²(2)=21.5 | <0.001* | χ²(2)=13.2 | 0.001* | χ²(2)=7.3 | 0.03 | χ²(2)=4.4 | 0.11 | χ²(2)=12.2 | 0.002* | χ²(2)=8.0 | 0.02 | χ²(2)=24.3 | <0.001* | χ²(2)=11.8 | 0.003 |

*P value<0.002.
†n=497 (99.8% valid responses).
‡n=462 (92.8% valid responses).
§Adjusting for all socio-demographic characteristics.
¶n=355 (99.7% valid responses).
**n=337 (94.7% valid responses).
LFT, lateral flow test.

might this be? We have previously shown that use of a test is more common in employees who work for specific sectors where daily LFTs are the norm.[13] It may be that taking a test is a marker of working in a sector where tackling the spread of infectious disease is viewed as a priority. It may also be that using tests indicates adherence to government advice in general.

We found relatively few associations between sociodemographic variables and contact behaviours when symptomatic, suggesting that these behaviours are determined by factors in common across the UK. Three exceptions stand out. First, participants with dependent children were more likely to leave home when ill in order to care for a vulnerable person. We speculate that having dependent children at home may also be associated with having elderly relatives (eg, grandparents) who may require care. Second, participants who reported greater financial hardship were less likely to leave home to care for vulnerable people. We are not clear why this is. Third, participants in later survey waves were generally less likely to report leaving home than those in the first wave. At the time of our data collection, levels of contact behaviours outside of the home were relatively stable across the population.[14] It is possible that our finding is because the first wave of our data collection coincided with the start of a new academic year, with parents and students feeling more obliged to leave home that week than normal.

Throughout the COVID-19 pandemic, a number of studies have focused on whether people who have COVID-19 adhere to advice to self-isolate. This work has identified a range of factors that may reduce adherence, including the absence of symptoms,[15] having alternative explanations for symptoms,[16] financial challenges[17] and the emotional impact of isolation.[18] However, the context of these studies makes it challenging to generalise from them to the issue of staying at home for a non-COVID-19 infectious illness. For example, in the UK at the time of our data collection people who received a positive COVID-19 test result were legally obliged to self-isolate, were eligible for statutory sick pay from the first day of their illness, as well as an additional support payment if they were on a low income, received regular support calls and text messages from a national service throughout their self-isolation, and could access a range of support packages including food deliveries, dog walking and other services depending on provision in their local area. In contrast, people with influenza-like illness who did not have a positive COVID-19 test result were only eligible for statutory sick pay starting from the fourth day of their illness; no additional support was provided.

As part of the UK government's plan to 'live with' SARS-CoV-2, advice from ministers has been for members of the public to treat COVID-19 in the same way as 'all other infectious diseases'.[19] As testing and support mechanisms for COVID-19 are withdrawn, identifying approaches to reduce the impact of undifferentiated acute respiratory illnesses will become ever more important. To our knowledge, there have been no randomised controlled trials or natural experiments evaluating potential approaches to reduce attendance at work by people with an infectious illness. This is likely to be a fruitful avenue of research for future studies.

## Strengths and limitations

The use of a large sample (n=8547) allowed us to identify a reasonably large subset of people who had experienced recent symptoms in the absence of a positive COVID-19 test result (n=498). By asking this group about behaviours that had occurred relatively recently, we reduced the impact of recall bias. Nonetheless, our study has several limitations. First, although we used standard market research practice to obtain our sample, we cannot be sure about its representativeness. Second, we asked people if they had developed symptoms in the past 10 days, and what actions they had taken after developing symptoms. If symptoms developed 10 days ago and actions were taken yesterday, the participant might have been outside their infectious window. However, official advice is that people remain infectious with a common cold for 2 weeks and with influenza for 8 days, suggesting this was not a major limitation.[20] Third, we relied on self-report of behaviour. Given that attending work, socialising and caring for vulnerable people while symptomatic may be socially undesirable to admit, the rates we identified for these outcomes are likely to be underestimates.

## CONCLUSION

Substantial numbers of people who have symptoms of an infectious illness attend work, go shopping, socialise with others and provide care to vulnerable people. This occurs at a high rate even when people suspect that they may be infectious. As SARS-CoV-2 adds to the burden of infectious diseases on populations, a greater focus on developing and evaluating ways to reduce presenteeism is more important than ever.

**Author affiliations**
[1]Department of Psychological Medicine, King's College London, London, UK
[2]UK Health Security Agency, Porton Down, London, UK
[3]ADMMH, King's College London, London, UK
[4]Institute of Health Informatics, University College London, London, UK
[5]Centre for Behaviour Change, University College London, London, UK

**Acknowledgements** We are grateful to all participants for giving up their time for this study, and to the English Department of Health and Social Care for making these data available to us.

**Contributors** GJR, RA, NTF, HP, SM and LES conceptualised the study. LES carried out formal analyses. GJR wrote the first draft of the manuscript and is guarantor. RA, NTF, HP, SM and LES reviewed and edited the manuscript. All authors approved the final version.

**Funding** This work was funded by the National Institute for Health Research (NIHR) Health Services and Delivery Research programme (NIHR project reference number (11/46/21)). Surveys were commissioned and funded by Department of Health and Social Care (DHSC), with the authors providing advice on the question design and selection. LES, RA and GJR are supported by the National Institute for Health Research Health Protection Research Unit (NIHR HPRU) in Emergency Preparedness and Response, a partnership between the UK Health Security Agency, King's College London and the University of East Anglia. RA is also supported by the NIHR HPRU in Behavioural Science and Evaluation, a partnership between the UK Health Security

Agency and the University of Bristol. HP has received funding from Public Health England and NHS England. NTF is part funded by a grant from the UK Ministry of Defence. The views expressed are those of the authors and not necessarily those of the NIHR, Public Health England, the Department of Health and Social Care or the Ministry of Defence. The Department of Health and Social Care funded data collection (no grant number).

**Competing interests**  All authors had financial support from NIHR for the submitted work; RA is an employee of the UK Health Security Agency; HP has received additional salary support from Public Health England and NHS England; HP receives consultancy fees to his employer from Ipsos MORI and has a PhD student who works at and has fees paid by Astra Zeneca; at the time of writing GJR is acting as an expert witness in an unrelated case involving Bayer PLC, supported by LES; no other financial relationships with any organisations that might have an interest in the submitted work in the previous three years; no other relationships or activities that could appear to have influenced the submitted work. NTF is a participant of an independent group advising NHS Digital on the release of patient data. At the time of writing this paper, all authors were participants of the UK's Scientific Advisory Group for Emergencies or its subgroups.

**Patient and public involvement**  Patients and/or the public were involved in the design, or conduct, or reporting, or dissemination plans of this research. Refer to the Methods section for further details.

**Patient consent for publication**  Not applicable.

**Ethics approval**  This study involves human participants but this work was conducted as a service evaluation of the Department of Health and Social Care's public communications campaign and, following advice from King's College London Research Ethics Subcommittee, was exempt from ethical approval. Participants of online research panels have consented to being contacted to take part in online surveys. Following industry standards, consent was implied by participants' completion of the survey.

**Provenance and peer review**  Not commissioned; externally peer reviewed.

**Data availability statement**  No data are available.

**ORCID iDs**
Louise E Smith http://orcid.org/0000-0002-1277-2564
Nicola T Fear http://orcid.org/0000-0002-5792-2925

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
