## [Reviewer comments · BMJ Open]

ARTICLE DETAILS

TITLE (PROVISIONAL)	Do people with symptoms of an infectious illness follow advice to stay at home? Evidence from a series of cross-sectional surveys about presenteeism in the UK
AUTHORS	Rubin, G James; Smith, Louise; Amlot, Richard; Fear, Nicola; Potts, Henry; Michie, Susan

VERSION 1 – REVIEW

REVIEWER	Alison Bacon University of Plymouth, Psychology
REVIEW RETURNED	10-Feb-2022

GENERAL COMMENTS	Review of bmjopen-2021-060511: Do people with symptoms of an infectious illness follow advice to stay at home? Evidence from a series of cross-sectional surveys in the UK. A very clearly written and concise paper, which has many strengths. This is an important topic as presenteeism is a long-standing issue in terms of workplace health. There are however some areas where further detail would be useful. 1. How the participants were sampled. Why were these particular survey waves chosen and were these question asked at other times? If so, why this particular timespan.2. Results section, paragraph 2 states that most participants reported having been to work after developing their symptoms, and quotes relevant percentages. However, some participants in the sample were not working. This part of the report needs clarification to show whether this is percentage of participants who reported being employed, or of the overall sample. I think the former may be more informative, especially given the pressure on many employees to attend work even when unwell (as noted by the authors in the discussion).3. Was there a difference in other forms of going out between workers and non-workers? People in employment may think as they are going to work, they may as well go other places as well. Do we have data on why people were not employed, some may be on long-term sick, for instance, and that might be the main driver of their behaviour.4. Is there any data on the type of employment? A substantial percentage of the population are unable to work at home because of the nature of their job. This may also affect their decision. If the data do not allow for address of pints 2-4 above, then the authors could usefully mention these outstanding issues in the discussion.
---

REVIEWER	bisakha sen University of Alabama at Birmingham, Healthcare Organization & Policy
REVIEW RETURNED	27-Feb-2022

GENERAL COMMENTS	BMJ Open Review This is a moderately interesting study that provides descriptive results for how individuals in UK participated in behaviors that might lead to spread of infection even though they had symptoms indicative of COVID-19, and in spite of government level warnings to stay home. The manuscript could be strengthened in several ways, by using more complete data, by putting more context around their somewhat simplistic findings and by using a richer statistical approach. That said, I realize the authors are restricted to the information that has already been collected via survey. 1) Members from this team of authors have published several papers based on the surveys used here (they have used different numbers of surveys in different papers). To assure readers about lack of redundancy, it would help if this paper could be placed in context of the several other publications they have on adherence/non-adherence to isolation during COVID19 (e.g. Adherence to the test, trace and isolate system: results from a time series of 21 nationally representative surveys in the UK (the COVID-19 Rapid Survey of Adherence to Interventions and Responses [CORSAIR] study) 2) The data used came from 20 September 2021 to 3 November 2021 surveys (as per abstract). Since the pandemic was ongoing for most of 2020 and 2021, why did the authors only focus on 4 surveys (whereas in their cited work (ref 6) they used 37 surveys) ? The authors are strongly encouraged to use additional surveys to give a more complete picture. 3) In the abstract and in the manuscript, authors should desist from use of sentences like “most people are not” – this is an emotional, and not well-balanced, way of describing results. For some results – e.g. 51.2% in case of grocery shopping – “approximately half” or “a little more than half” is more accurate. Other behaviors like caring for a vulnerable person were undertaken by a smaller minority (40% or so). 4) The literature review is inadequate. There is a literature on COVID19 isolation compliance that should be cited (e.g. Steens et al for Norway, Bodas et al, Israel, Carluchhi et al, Italy). 5) It is likely that, by Sept 2021, a substantial percent of the UL population had already had COVID19. Did the survey ask people if they had already had COVID19 ? Presumably that could be a major reason for people believing that they were not infectious. 6) Apart from race/nationality, did the survey collect any other indicators of socio-economic status? It would be interesting to know whether those “not following” government directives are more likely be affluent and white-collar workers or not. 7) Similarly, did the survey collect any information on age ? 8) In the introduction, authors make several references to government directives to stay home if people were experiencing symptoms. For readers not familiar with all UK government policies, it would be particularly useful to know a. What provisions were being made to support hourly or temporary workers who could not go to work ? Or did they simply lose their income for days when they could not work? b. What provisions, if any, were in place to ensure people with symptoms could get essentials like groceries, other essentials or
--

	pharmaceutical products delivered? Otherwise, these trips might have happened out of dire necessity, rather than flippant unconcern about safety. c. What provisions were put in place for caregiving of vulnerable persons if their family members could not provide it ? Otherwise that caregiving could also be driven by sheer necessity. d. Here is one example of a “suggested list” for quarantining in the US (https://www.cleveland.com/coronavirus/2020/03/22-essentials-for-self-quarantine-and-isolation.html). It is obvious even from a cursory glance that a family would have to have some degree of affluence and savings to buy most of these products based on a “just in case.....” 9) If the data are available in the survey, a richer statistical analyses that explores predictors of non-adherence to each behavior (perhaps a logit model) – that showed the role of gender, age, indicators of socio-econ status, employment status, prior COVID19 infection, believing their symptoms were due to COVID19, and any information on whether they had alternate resources to buy essentials or care for a vulnerable person instead of being compelled to do it themselves – would be of far greater use to policymakers and scientists. The current, mostly descriptive approach provides a very partial picture. Other suggestions Table 1: Please fix the answers under the “yes” for Go to Work column. A breakdown of how many people attributed symptoms to “hayfever / allergies,” “a cold,” “asthma,” “flu,” “coronavirus,” “long COVID,” “other, please specify” [with a free-text space provided] and “don’t know” would be useful. It is well-established that the loss of smell and taste was fairly unique to COVID19, while other symptoms could arise from multiple sources. What fraction of those who reported loss of smell/test thought their symptoms were from COVID19, and got tested ?
--	---

VERSION 1 – AUTHOR RESPONSE

Reviewer: 1

Dr. Alison Bacon, University of Plymouth

1. How the participants were sampled. Why were these particular survey waves chosen and were these question asked at other times? If so, why this particular timespan.

We have amended the relevant section of text to explain this. As this now states: “Participants were aged over 16 years, lived in the UK and had previously opted-in to receive invitations to take part in online market research surveys. Participants who took part in one wave of the Department of Health and Social Care series of surveys were prevented from taking part in any of the next three waves. We therefore elected to use the most recent four waves to ensure that each participant only contributed one set of response. The market research company sent invitations to existing panel members who met these criteria, asking them to complete the survey. Quotas were used based on age and gender (combined) and region to ensure the sample was broadly representative of the UK population on these variables. Quotas were based on mid-year (2018) projections from the Office for National Statistics.”

2. Results section, paragraph 2 states that most participants reported having been to work after developing their symptoms, and quotes relevant percentages. However, some participants in the sample were not working. This part of the report needs clarification to show whether this is

percentage of participants who reported being employed, or of the overall sample. I think the former may be more informative, especially given the pressure on many employees to attend work even when unwell (as noted by the authors in the discussion).

This analysis was indeed restricted to employed participants. We have clarified this in the relevant section of the results (“Most employed people…”).

3. Was there a difference in other forms of going out between workers and non-workers? People in employment may think as they are going to work, they may as well go other places as well. Do we have data on why people were not employed, some may be on long-term sick, for instance, and that might be the main driver of their behaviour.

We have added an analysis of socio-demographic predictors which includes workers and non-workers. While we do have data that could be used to split non-employed status into subcategories, the numbers start to get quite small and we have therefore not presented this.

4. Is there any data on the type of employment? A substantial percentage of the population are unable to work at home because of the nature of their job. This may also affect their decision. If the data do not allow for address of points 2-4 above, then the authors could usefully mention these outstanding issues in the discussion.

We have included “able to work from home” as one the sociodemographic variables in our new analysis.

Reviewer: 2

Dr. Bisakha Sen, University of Alabama at Birmingham

BMJ Open Review

1) Members from this team of authors have published several papers based on the surveys used here (they have used different numbers of surveys in different papers). To assure readers about lack of redundancy, it would help if this paper could be placed in context of the several other publications they have on adherence/non-adherence to isolation during COVID19 (e.g. Adherence to the test, trace and isolate system: results from a time series of 21 nationally representative surveys in the UK (the COVID-19 Rapid Survey of Adherence to Interventions and Responses [CORSAIR] study)

All of our papers from CORSAIR refer to the survey wave numbers of included data, allowing readers to assess where overlap occurs. We have now provided a reference to the study website where all reports, pre-prints and papers arising from this work are presented. We have also cited some of our previous work in the discussion section.

2) The data used came from 20 September 2021 to 3 November 2021 surveys (as per abstract). Since the pandemic was ongoing for most of 2020 and 2021, why did the authors only focus on 4 surveys (whereas in their cited work (ref 6) they used 37 surveys) ? The authors are strongly encouraged to use additional surveys to give a more complete picture.

The context within the UK has changed repeatedly during the pandemic. These contextual changes are important – for example, studying presenteeism during a period in which ‘stay at home’ orders were in force is very different to studying the phenomenon during a period of fewer restrictions. The

four survey waves we selected for analysis are appropriate because a) restricting the sample to only four waves helps to ensure that individual participants provided only a single response in the dataset (see our answer to Reviewer 1, point 1) and b) at the time the waves occurred, levels of behaviour in terms of leaving the home were relatively stable across the population (see our reference 14).

3) In the abstract and in the manuscript, authors should desist from use of sentences like “most people are not” – this is an emotional, and not well-balanced, way of describing results. For some results – e.g. 51.2% in case of grocery shopping – “approximately half” or “a little more than half” is more accurate. Other behaviors like caring for a vulnerable person were undertaken by a smaller minority (40% or so).

We have made these changes.

4) The literature review is inadequate. There is a literature on COVID19 isolation compliance that should be cited (e.g. Steens et al for Norway, Bodas et al, Israel, Carluchhi et al, Italy).

Bodas et al. assessed intentions to comply with a request to self-isolation. Given how poorly intentions map onto self-reported behaviour in this context [see, e.g., reference 6 in the manuscript], we have not discussed their paper. Carluchhi et al. assess adherence to general lockdown measures, rather than self-isolation – this has limited relevance for our study and we have not discussed it. However, we have now discussed Steens et al., as well as several other studies that have documented the challenges associated with self-isolating, both in the context of COVID-19 and for acute respiratory infections more generally.

5) It is likely that, by Sept 2021, a substantial percent of the UL population had already had COVID19. Did the survey ask people if they had already had COVID19 ? Presumably that could be a major reason for people believing that they were not infectious.

We have now included this in our analyses – there is no impact on our outcome measures.

6) Apart from race/nationality, did the survey collect any other indicators of socio-economic status? It would be interesting to know whether those “not following” government directives are more likely be affluent and white-collar workers or not.

We have now provided these analyses.

7) Similarly, did the survey collect any information on age ?

We have now provided these analyses.

8) In the introduction, authors make several references to government directives to stay home if people were experiencing symptoms. For readers not familiar with all UK government policies, it would be particularly useful to know

a. What provisions were being made to support hourly or temporary workers who could not go to work ? Or did they simply lose their income for days when they could not work?

- b. What provisions, if any, were in place to ensure people with symptoms could get essentials like groceries, other essentials or pharmaceutical products delivered? Otherwise, these trips might have happened out of dire necessity, rather than flippant unconcern about safety.
- c. What provisions were put in place for caregiving of vulnerable persons if their family members could not provide it? Otherwise that caregiving could also be driven by sheer necessity.
- d. Here is one example of a “suggested list” for quarantining in the US (<https://www.cleveland.com/coronavirus/2020/03/22-essentials-for-self-quarantineand-isolation.html>). It is obvious even from a cursory glance that a family would have to have some degree of affluence and savings to buy most of these products based on a “just in case.....”

We have expanded our discussion section to provide greater detail on this and in particular to note the disparity between the support provided to people with a positive COVID-19 test result, and the lack of support provided to people who are symptomatic but do not have a COVID-19 test result (and who represent the population of interest for this paper).

9) If the data are available in the survey, a richer statistical analyses that explores predictors of non-adherence to each behavior (perhaps a logit model) – that showed the role of gender, age, indicators of socio-econ status, employment status, prior COVID19 infection, believing their symptoms were due to COVID19, and any information on whether they had alternate resources to buy essentials or care for a vulnerable person instead of being compelled to do it themselves – would be of far greater use to policymakers and scientists. The current, mostly descriptive approach provides a very partial picture.

We have now provided a fuller account of the association between sociodemographic variables and behaviours.

Other suggestions

Table 1: Please fix the answers under the “yes” for Go to Work column.

We are not clear what needs fixing here. Perhaps some formatting issue crept in when the file was converted by the journal’s online system? We will check with the editorial team if the paper is accepted.

A breakdown of how many people attributed symptoms to “hayfever / allergies,” “a cold,” “asthma,” “flu,” “coronavirus,” “long COVID,” “other, please specify” [with a free-text space provided] and “don’t know” would be useful.

We have provided this.

It is well-established that the loss of smell and taste was fairly unique to COVID19, while other symptoms could arise from multiple sources. What fraction of those who reported loss of smell/test thought their symptoms were from COVID19, and got tested?

This is outside the remit of the current paper. Previous papers by our group have assessed factors associated with uptake of testing. See e.g. references 5 and 13 in the manuscript.

VERSION 2 – REVIEW

REVIEWER	bisakha sen University of Alabama at Birmingham, Healthcare Organization & Policy
REVIEW RETURNED	13-May-2022
GENERAL COMMENTS	The authors have addressed most comments adequately. While I still believe the paper makes a relatively minor contribution given that the authors have published similar papers from previous surveys, and while some limitations in the survey itself prevent them from answering some really informative questions, I am okay with the manuscript being accepted for publication.